# Sex Difference on Neurological Outcomes and Post-Cardiac Arrest Care in Out-of-Hospital Cardiac Arrest Patients Treated with Targeted Temperature Management: Post-Hoc Study of a Prospective, Multicenter, Observational Cohort Study

**DOI:** 10.3390/jcm12165297

**Published:** 2023-08-14

**Authors:** Seon Yeong Park, Sang Hoon Oh, Sang Hyun Park, Jae Hun Oh, Soo Hyun Kim

**Affiliations:** 1Department of Emergency Medicine, Yeouido St. Mary’s Hospital, College of Medicine, The Catholic University of Korea, Seoul 07345, Republic of Korea; embracesy@gmail.com (S.Y.P.); emergency70@hanmail.net (S.H.P.); 2Department of Emergency Medicine, Seoul St. Mary’s Hospital, College of Medicine, The Catholic University of Korea, Seoul 06591, Republic of Korea; mdshow@hanmail.net; 3Department of Emergency, Eunpyeong St. Mary’s Hospital, College of Medicine, The Catholic University of Korea, Seoul 03312, Republic of Korea; emojh@catholic.ac.kr

**Keywords:** out-of-hospital cardiac arrest, sex, in-hospital course, targeted temperature management

## Abstract

Conflicting results regarding sex-based differences in the outcomes of out-of-hospital cardiac arrest (OHCA) patients have been reported. We aimed to evaluate the association between sex and neurological outcome as well as various in-hospital process in OHCA patients treated with targeted temperature management. We retrospectively analyzed a prospective registry data collected between October 2015 and December 2018. To evaluate the effect of sex on patient outcomes, we created various multivariable logistic regression models. When the results were adjusted using resuscitation variables and in-hospital variables, there was no significant difference (OR = 1.22, 95% CI: 0.85–1.74; OR = 1.13, 95 CI: 0.76–1.68, respectively). Regarding the in-hospital course, the daily total SOFA score was similar in both sexes, whereas cardiovascular scores were higher in women on days 2 and 3. The adjusted effect of sex was not associated with the clinician’s decision to perform early cardiac interventions, except for those men that had more extracorporeal membrane oxygenation (OR = 2.51, 95% CI: 1.11–5.66). The findings seems that men had more favorable 6-month neurological outcomes. However, after adjusting for confounders, there was no difference between the sexes. The results regarding in-hospital course were similar in men and women.

## 1. Introduction

Sex-based differences regarding disease arise from genetic differences between men and women [1], as well as the potential for bias causing differences in patients care and ultimately their outcomes. In particular, in cardiovascular disease, sex differences in symptoms and underlying pathophysiology have been increasingly recognized [2,3], and have been used to improve health care. However, sex differences in survivors after out-of-hospital cardiac arrest (OHCA) remain unclear despite an increasing number of studies and the substantial public health burden [4].

Previous reports regarding the association between sex and outcome after OHCA have been conducted in heterogeneous populations, with the majority based on emergency medical service-attended OHCA [5,6,7], and others conducted on those who survived and were admitted to a hospital [8,9,10]. Consequently, the majority have mainly focused on outcomes at hospital discharge and the results regarding the association between sex and outcomes are conflicting. Several studies showed better outcomes in men or women [5,6], some demonstrated comparable outcomes [7,8,9,10], and other studies reported better outcomes in women of reproductive age [11]. Even recently published meta-analyses have presented conflicting conclusions [12,13,14,15]. Post-cardiac arrest syndrome after OHCA is a heterogeneous disease entity with various causes and severities, and the in-hospital processes including diagnostic and therapeutic interventions that follow are also diverse. However, most previous reports could not present the utilization difference of such in-hospital interventions between sexes [16], which affect in-hospital interim outcomes and eventually, long-term neurological outcomes. Therefore, discrepancies in the outcomes among the published studies may be explained by different patient selection or incomplete analysis [6]. Furthermore, in-hospital death in OHCA patients usually results from the withdrawal of life-sustaining therapy (WLST) with or without a waiting period for neurological improvement [17]. While men and women may differ in their preference for life-sustaining treatments (LSTs) as a function of the specific type of illness for which LSTs are considered [18], limited studies have examined the differences in decisions regarding end-of-life in survivors of OHCA by sex [19].

In this study, we used a large prospective registry that recorded the details of post-cardiac arrest care and evaluated the sex difference in comatose OHCA patients treated with targeted temperature management (TTM). Our primary aim was to evaluate the adjusted association between sex and long-term neurological outcomes. The secondary aim was to investigate the difference in in-hospital processes between men and women by comparing changes in daily Sequential Organ Failure Assessment (SOFA) scores, utilization of therapeutic interventions and diagnostic tests, and decisions regarding limiting active treatment (LAT) during hospitalization.

## 2. Materials and Methods

### 2.1. Study Design and Setting

This was a post-hoc study of a prospective, multicenter, observational cohort study based on the Korean Hypothermia Network Prospective Registry 1.0 (KORHN-PRO 1.0), which is a web-based registry of OHCA patients treated with TTM. The KORHN-PRO 1.0 protocol was approved by the Institutional Review Board of each hospital and was registered in the clinical trial registry platform (NCT02827422). Details on the KORHN-PRO 1.0 study design, protocol, and results have been published previously [20]. Informed consent was obtained from the patients’ legal relatives.

### 2.2. Study Population

From October 2015 to December 2018, 22 participating hospitals enrolled 1373 adult OHCA patients (≥18 years old) who were unconscious after the return of spontaneous circulation (ROSC) and treated with TTM. The current study included all adult patients treated with TTM. The exclusion criterion was missing data on neurological outcomes after 6 months.

### 2.3. Measurements

All variables were extracted from the KORHN-PRO 1.0 registry. Patient baseline characteristics, including sex information, age, and comorbidities, were collected. Each comorbidity was integrated into a modified comorbidity index (mCI) value based on the Charlson comorbidity index (CCI), which was validated as the degree of comorbidity in these patients by Winther et al. [21].

The following resuscitation variables were collected: the location of the arrest, whether the arrest was witnessed, bystander cardiopulmonary resuscitation (CPR), initial arrest rhythm (shockable versus non-shockable), arrest etiology (presumed cardiac versus non-cardiac), and downtime (duration from arrest time to ROSC), as well as Glasgow Coma Scale (GCS) motor grade, brainstem reflexes (both pupillary light reflex and corneal reflex) (both absence or not), and electrocardiogram (ECG) (ST-segment elevation myocardial infarction [STEMI]/LBBB [left bundle branch block] or not) immediately after ROSC.

As in-hospital variables, we investigated cardiac interventions (early coronary angiography [CAG], percutaneous coronary intervention [PCI], intra-aortic balloon pumping [IABP] and echocardiography, and extracorporeal membrane oxygenation [ECMO]). Early intervention was defined as that performed within 24 h after ROSC. Whether the patients underwent the following prognostic tests was also determined: brainstem reflexes on day 4, brain computed tomography (CT) or diffusion-weighted image (DWI), electroencephalography (EEG), serum level of neuron-specific enolase (NSE), and somatosensory evoked potential (SSEP). The prognosis was considered poor when there was one abnormal result for any single predictor (the absence of brainstem reflexes, injury on brain CT or DWI, highly malignant EEG, high NSE level, and absence of the N20 wave on SSEP). We also used the multimodal algorithm recommended by the European Resuscitation Council/European Society of Intensive Care Medicine considering a combination of ≥2 abnormal results on any prognostic tests to indicate a poor outcome [22,23].

The Daily Sequential Organ Failure Assessment (SOFA) score and six sub-scores (respiration, coagulation, liver, cardiovascular, central natural system, and renal) were assessed as interim outcomes [24]. We assessed whether there was any decision regarding LAT, which consisted of WLST, withholding LST, and do-not-attempt CPR (DNAR) orders, and relative’s refusal of further diagnostic tests was pre-defined as original KORHN-PRO variables.

### 2.4. Statistical Analysis

The categorical variables are presented as the total number of patients and the proportion of patients, and the continuous variables are reported as the mean ± standard deviation. To compare patient characteristics between the two groups, we used the chi-squared test and the Student’s *t*-test.

To evaluate the effect of sex differences on the primary outcome, logistic regression models were used to assess the factors independently associated with good neurological outcomes. Baseline characteristics, resuscitation, and in-hospital factors were sequentially adjusted in each model to evaluate the association between sex and the outcomes. After univariate analysis, the variables of interest or those with a possible predictive value (*p* < 0.05) were entered into a multivariable logistic regression analysis, and odds ratios (ORs) and 95% confidence intervals (CIs) were calculated. To determine whether there was a difference according to age, an interaction term (age × sex) was included in the multivariable logistic regression model, and the two sexes were compared by dividing them into groups aged less than 50 and more than 50 years. To investigate the association between sex and in-hospital processes or interim outcomes, multivariable logistic regression was also performed using covariates. We used repeated measured analysis of covariance (ANCOVA) for the adjustment covariates to evaluate the effect of sex differences on the daily total SOFA scores and six sub-scores.

All statistical analyses were performed using IBM SPSS version 24 software (IBM, Armonk, NY, USA). All *p*-values were two-tailed, and *p* < 0.05 was considered significant.

## 3. Results

Of 10,258 patients experiencing OHCA during the study period, 1373 were registered in the KORHN-PRO registry. Among these, 34 were excluded from the analysis due to missing 6-month neurological outcomes. Of the remaining 1339 patients, 952 (71.1%) were men and 387 (28.9%) were women. After 6 months, 412 (30.8%) had good neurological outcomes and the other 927 (69.2%) had poor outcomes (Figure 1).

### 3.1. Comparison of Patient Variables between Men and Women

Table 1 shows the characteristics of the study participants according to sex. The mean age was similar between both sexes (*p* = 0.433). A history of acute myocardial infarction (AMI) was more common in men (7.6% vs. 3.6%, *p* = 0.008), and congestive heart failure (CHF) and chronic kidney disease (CKD) were more common in women (2.9% vs. 5.9%, *p* = 0.009; 6.5% vs. 10.3%, *p* = 0.017, respectively). However, in the analysis using mCI values, the comorbidity burden was similar between the sexes. In the arrest situation, men were more likely to experience witnessed cardiac arrest (72.3% vs. 62.3%, *p* < 0.001) and in places other than at home (48.0% vs. 59.4%, *p* < 0.001). The likelihood of shockable rhythm and cardiac etiology arrest was significantly higher in men (40.1% vs. 24.3%, *p* < 0.001; 65.8% vs. 52.5%, *p* < 0.001, respectively). STEMI or LBBB was more common in men (16.9% vs. 9.0%, *p* < 0.001), and early CAG and PCI, and ECMO were more frequently performed in men (31.0% vs. 19.8%, *p* < 0.001; 18.4% vs. 7.5%, *p* < 0.001; 5.8% vs. 1.8%, *p* = 0.002, respectively). There were no differences between the sex groups in the rate of prognostic tests and other interventions. The rate of decisions regarding LATs was similar in both sexes. The most common reason for limiting LST was DNAR orders in both sexes (165 cases [17.3%] in men and 71 cases [18.3%] in women), followed by withholding LST (*n* = 26), WLST (*n* = 12), and relative’s refusal of further diagnostic tests (*n* = 9). Finally, the survival rate to hospital discharge was not different, but the rate of good neurological outcomes after 6 months was higher in men (33.7% vs. 23.5%, *p* < 0.001).

### 3.2. Association between Sex and 6-Month Good Neurological Outcome

Table 2 presents the association between sex and neurological outcome. The details of the variables entered into each model are presented in Table 3. The 6-month neurological outcomes seemed to be better in men before adjustment (OR = 1.66, 95% CI: 1.26–2.17). This association was still significant after adjusting for pre-arrest variables (age and mCI) in model 1 (adjusted OR [AOR] = 1.78, 95% CI: 1.34–2.35). However, when the results were adjusted using resuscitation variables (model 2) and in-hospital variables were added (model 3), there was no significant difference between the two sexes (AOR = 1.22, 95% CI: 0.85–1.74; AOR = 1.10, 95 CI: 0.74–1.64, respectively).

The interaction between sex and age in model including all variables (model 3) was not significant for good neurological outcomes (*p* = 0.401) (Table 3). In a stratified analysis according to different age groups, the probability of better 6-month neurological outcomes in men was higher in both age groups (age < 50, OR = 1.80, 95% CI: 1.15–2.80, for age ≥ 50, OR = 1.72, 95% CI: 1.21–2.44), but this was not significant after adjusting for all covariates (AOR = 1.49, 95% CI: 0.72–3.06, *p* = 0.279; OR = 1.05, 95% CI: 0.62–1.76, *p* = 0.866, respectively) (Table 4).

### 3.3. Association between Sex and In-Hospital Processes or Organ Failure

The details of associating factors for early cardiac intervention are presented in Appendix A. Among 829 patients with presumed cardiac etiology arrests, men more commonly underwent early CAG (OR = 1.61, 95% CI: 1.16–2.24) according to the univariate analysis results. However, after adjusting for confounders, the odds of male sex was not significant (AOR = 1.18, 95% CI: 0.82–1.69). In patients who underwent early CAG (*n* = 368), PCI was more frequently performed in men (OR = 1.88, 95% CI: 1.10–3.21). However, this association was not significant when adjusting for confounders (AOR = 1.25, 95% CI: 0.56–2.80). Early echocardiography was not associated with a specific sex (AOR = 1.02, 95% CI: 0.77–1.35, respectively). In contrast, the AOR of men for ECMO was 2.51 (95% CI: 1.11–5.66).

In total, 672 patients underwent multimodal prognostic tests. Of these, we assessed the independent predictors for any decisions regarding LAT in two models that included each single predictor and multimodal prognostication according to international guideline, respectively (Table 5). While older age, the absence of brainstem reflex and SSEP response, or abnormal results in more than two predictors were associated with decisions regarding LAT, the sex had no association with these decisions in either model (AOR = 1.20, 95% CI: 0.77–1.88, OR = 1.17, 95% CI: 0.76–1.82, respectively).

Daily total SOFA scores tended to gradually decrease in both sexes, and there was no difference between the sexes. In the analysis of the six sub-scores, the mean values of several sub-scores were different between the sexes (Figure 2). After adjusting for confounders such as age, comorbidities (mCI value), shockable rhythm, arrest time, cardiac etiology arrest, and STEMI/LBBB, we found that cardiovascular sub-scores on hospital days 2 and 3 were higher in women than those in men (on day 2, *p* = 0.006; on day 3, *p* = 0.017). Nonetheless, liver and renal sub-scores were higher in men than in women 3 days (all *p*-values < 0.05).

## 4. Discussion

In this multicenter registry-based study, we evaluated the sex differences in long-term neurological outcomes, in-hospital processes including diagnostic and therapeutic interventions, and daily SOFA score among adult OHCA patients treated with TTM. Although in the univariate analysis it seemed that men had better 6-month neurological outcomes, when adjusted for confounders, the difference was not statistically significant. Prognostic tests and cardiac interventions except for ECMO and decisions regarding LAT were not different between the sexes. In contrast, we found sex differences in the daily injury severity scores of cardiovascular, liver, and renal organs.

Our final findings are consistent with those of previous studies and strengthen the evidence that sex has no association with neurological outcomes in TTM patients after OHCA [8,9,10]. Although overall, men were significantly more likely to have a good neurological outcome after 6 months than women, there were several differences in pre-arrest, intra-arrest, and post-arrest factors between the sexes. Men had more AMI as comorbidity, whereas women had more CHF and CKD. Men seemed to have a more favorable resuscitation factor than women. The sex difference in neurological outcomes remained after adjusting for demographic variables. However, after adjusting for resuscitation and in-hospital confounding variables, sex was not associated with neurological outcomes in these patients. Several hypotheses could explain the conflicting results regarding the association between sex and outcomes following OHCA in previous studies [5,6,7,8,9,10,11,19]. Our study differed from most other studies in that analysis based on in-depth hospital registry, which allowed us to adjust for in-hospital factors. Men may have advantages not only in a resuscitation situation, but also in hospital care, so different results could be obtained in studies that do not control for these variables.

Interestingly, hospital-based studies and those only including patients treated with TTM consistently demonstrated similar outcomes between sexes [8,9,10].

Conversely, there is an argument that women may have advantages in cardiac arrest due to the neuroprotective effects of estrogen [25,26]. Some clinical studies also demonstrated that women had good neurological outcomes compared to men, especially at reproductive ages [11]. One explanation might be that the neuroprotective effect of endogenous estrogen is attenuated or underestimated in the human TTM setting using clinical measures of neurological function. This can be partially explained that studies in which women have better outcome than men had lower TTM application rates [27]. Another possible cause is the age difference of the patients enrolled in the studies. However, our cohort was the youngest in the published literature, and sex did not influence the neurological outcomes in the stratified analysis or in interaction analysis with age. Although we tried to adjust identifiable confounders between the sexes, there is the possibility of implicit or explicit bias by sex, which are sometimes unavoidable and uncontrolled.

To understand sex-based differences in these patients, we tried to investigate the in-hospital processes as well as neurological outcome. First, we evaluated the rates of cardiac diagnostic and therapeutic interventions in both sexes. While coronary catheterization is currently used as the standard of care for post-cardiac arrest, previous studies showed that the adjusted rates of early CAG were significantly lower in women than in men [8,28,29]. In this study, early CAG and PCI were likely to be underutilized in women. However, these were explained by higher proportions of shockable rhythm, ECG findings, or more severe obstruction findings on CAG. There was no difference in the utilization of echocardiography between both sexes, and a higher proportion of men underwent ECMO. Considering that women had more CHF before OHCA and cardiovascular instability during early hospital care, different management strategies, including early echocardiography and ECMO, could be helpful for women.

Second, health disparities can be affected by differences in resource consumption according to social determinants such as sex. However, in our results, men and women did not experience inequity with respect to prognostic evaluations.

Third, currently, little information is available regarding the rates of WLST and limiting active treatment in men and women [19]. Some previous studies, which identified that women were less likely to survive to hospital discharge after OHCA, postulated that this was due to increased WLST [30,31]. We included the results of prognostic test to analyze the association between sex and end-of-life decisions. Although only small proportion of our cohort had WLST, women had rates of WLST and other LAT similar to those of men, which is in contrast to some previous studies but not others [8,19,32]. The increase in age and the absence of brainstem reflex and an N20 peak on SSEP (or >2 poor outcome predictors) were independently associated with these decisions but not sex.

Lastly, we analyzed daily total SOFA scores and six sub-scores between the sexes. To our knowledge, the present study was the first clinical report to describe the sex-based difference on daily patient severity using organ severity scores. Several findings are interesting to clinicians. In contrast to total SOFA scores, several sub-scores differed between the sexes for 3 days after ROSC. In particular, cardiovascular instability on days 2 and 3 in women was observed, and they had the wide range of cardiovascular sub-scores compared to other sub-scores.

There were a number of limitations to this study. First, although our study included more in-depth investigation from pre-arrest to 6 months after ROSC, the nature of a retrospective analysis inevitably leads to selection bias, information bias, and other confounders that are unavoidable. Second, some studies reported that women usually have a higher rate of ROSC after OHCA considering unfavorable resuscitation factors [33,34], and the rate of achieving ROSC from OHCA is an important interim outcome in understanding sex differences in OHCA, and consequently, the impact on long-term neurological outcomes. However, our registry is limited to patients treated with TTM after ROSC and rate of ROSC did not include these details. Thus, our results cannot be extended to entire OHCA patients, and the findings must be cautiously interpreted in comparison to other studies. Third, our data included few deaths after WLST, which was the most common mode of death in Western studies. Therefore, the cultural context of South Korea should be considered in interpreting our results regarding the effect of sex differences on end-of-life decisions. Fourth, healthcare costs, which could be an obstacle to appropriate in-hospital processes, were not considered in this study. Nonetheless, we tried to adjust for the identifiable differences in pre-arrest, intra-arrest characteristics, and all in-hospital processes.

## 5. Conclusions

Unadjusted, men were more likely to have good 6-month neurological outcomes. However, after adjustment for confounders, male sex was not associated with good neurological outcomes. The adjusted association between sex and in-hospital interventions or decisions regarding LAT was not significant except for the use of ECMO. Daily organ injury severity scores, especially cardiovascular, differed between the sexes after adjusting for covariates.

## Figures and Tables

**Figure 1 jcm-12-05297-f001:**
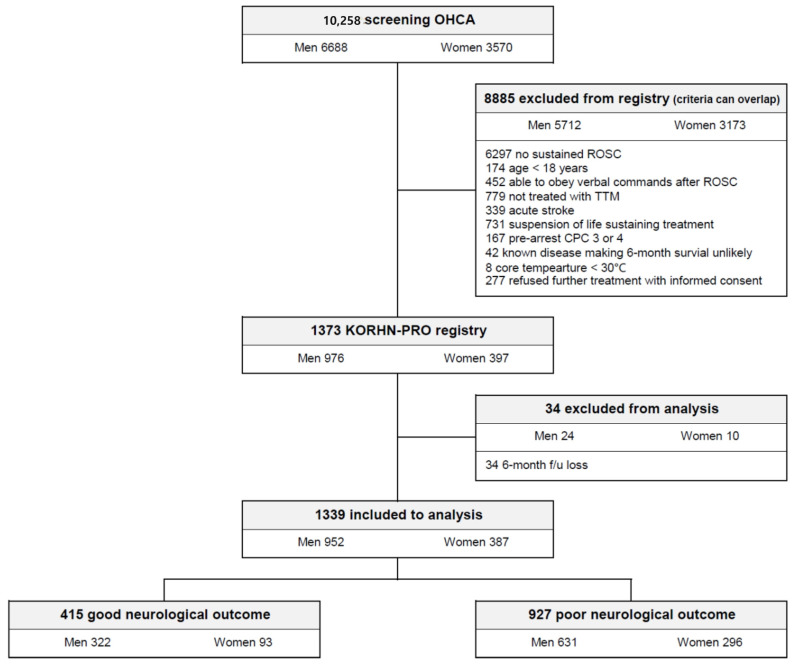
Flowchart of participant selection.

**Figure 2 jcm-12-05297-f002:**
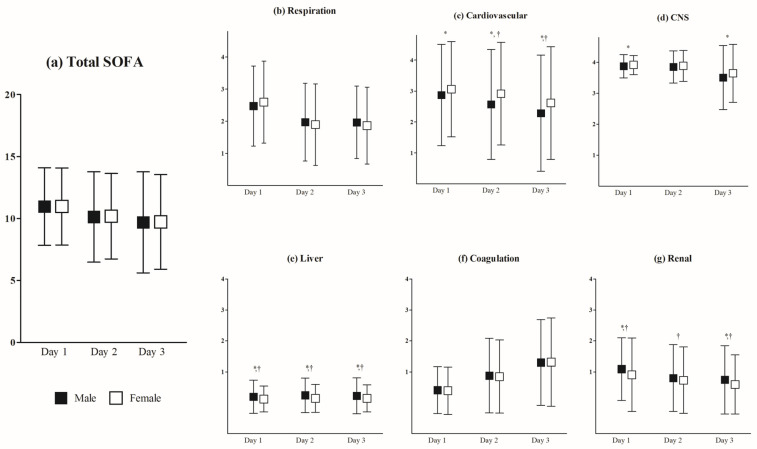
Differences in daily in-hospital SOFA scores between men and women. (**a**) Differences in the daily total SOFA score. (**b**) Differences in the daily sub-score of respiration SOFA, (**c**) coagulation SOFA, (**d**) liver SOFA, (**e**) cardiovascular SOFA, (**f**) CNS SOFA, and (**g**) renal SOFA. SOFA score curves show the means and standard deviations. * *p* < 0.05 in student *t*-test. † *p* < 0.05 in analysis of covariance. Abbreviations: SOFA, Sequential Organ Failure Assessment; CNS, central nervous system.

**Table 1 jcm-12-05297-t001:** Characteristics of the study participants by sex.

	Men (*n* = 952)	Women (*n* = 387)	*p*
Baseline variables			
Age, years	58.2 ± 15.2	57.5 ± 17.0	0.433
Comorbidity			
Acute myocardial infarction	72 (7.6)	14 (3.6)	0.008
Angina pectoris	57 (6.0)	22 (5.7)	0.831
Congestive heart failure	28 (2.9)	23 (5.9)	0.009
Arrhythmia	49 (5.1)	16 (4.1)	0.434
Stroke	47 (4.9)	26 (6.7)	0.193
Hypertension	330 (34.7)	151 (39.0)	0.132
Diabetes mellitus	231 (24.3)	95 (24.5)	0.913
Chronic lung disease	63 (6.6)	24 (6.2)	0.779
Chronic renal disease	62 (6.5)	40 (10.3)	0.017
Liver cirrhosis	18 (1.9)	4 (1.0)	0.263
Malignancy	46 (4.8)	25 (6.5)	0.228
mCI			
mCI 0	541 (56.8)	227 (58.7)	0.540
mCI 1	223 (23.4)	74 (19.1)	0.086
mCI 2	101 (10.6)	44 (11.4)	0.685
mCI 3	87 (9.1)	42 (10.9)	0.335
Resuscitation variables			
Arrest at home	457 (48.0)	230 (59.4)	<0.001
Witnessed	688 (72.3)	241 (62.3)	<0.001
Bystander CPR	591 (62.1)	231 (59.7)	0.415
Presumed cardiac etiology	626 (65.8)	203 (52.5)	<0.001
Downtime, min *	27.0 ± 14.5	27.0 ± 14.9	0.894
GCS motor grade > 2	155 (16.4)	44 (11.4)	0.022
Brainstem reflex	162 (17.0)	54(14.0)	0.190
SOFA score on day 1			0.208
4–7	65 (6.8)	33 (8.5)	
8–12	313 (32.9)	140 (36.2)	
≥12	574 (60.3)	214 (55.3)	
STEMI or new onset LBBB	161 (16.9)	35 (9.0)	<0.001
In-hospital variables			
Early coronary angiography	295 (31.0)	73 (19.8)	<0.001
Early percutaneous coronary intervention	175 (18.4)	29 (7.5)	<0.001
Early echocardiography	520 (54.6)	192 (49.6)	0.096
Extracorporeal membrane oxygenation	55 (5.8)	7 (1.8)	0.002
Brain computed tomography	874 (91.8)	364 (94.1)	0.158
Brain diffusion-weighted imaging	458 (48.1)	191 (49.4)	0.679
Electroencephalography	487 (51.1)	204 (52.4)	0.656
Somatosensory evoked potential	181 (19.0)	81 (20.8)	0.423
Neuron-specific enolase	411 (43.1)	153 (39.3)	0.201
Electrophysiology study	24 (2.5)	12 (3.1)	0.552
Limiting active treatments	182 (19.1)	79 (20.4)	0.587
Withdrawal of LST	10 (1.1)	2 (0.5)	0.348
Withholding of LST	19 (2.0)	7 (1.8)	0.822
Do-not-attempt-CPR	165 (17.3)	71 (18.3)	0.659
Refusal of further diagnostic test	4 (0.4)	5 (1.3)	0.077
Survival discharge	520 (54.6)	203 (52.5)	0.506
6-month good neurological outcome ^†^	321 (33.7)	91 (23.5)	<0.001

* Arrest-to-ROSC interval. ^†^ Good neurological outcome is defined as cerebral performance category 1 or 2. Abbreviations: mCI, modified Charlson comorbidity index; CPR, cardiopulmonary resuscitation; GCS, Glasgow coma scale; TTM, targeted temperature management; SOFA, Sequential Organ Failure Assessment; STEMI, ST elevation myocardial infarction; LBBB, left bundle branch block; LST, life sustaining therapy.

**Table 2 jcm-12-05297-t002:** Odd ratio for male sex for 6-month good neurological outcome by logistic regression.

	Univariate		Model 1		Model 2		Model 3	
	OR (95% CI)	*p*	OR (95% CI)	*p*	OR (95% CI)	*p*	OR (95% CI)	*p*
Good neurological outcome	1.66 (1.26–2.17)	<0.001	1.78 (1.34–2.35)	<0.001	1.22 (0.85–1.74)	0.276	1.10 (0.74–1.64)	0.626

Model 1: adjusted by baseline characteristics including male sex, age, and modified Charlson comorbidity index. For a survival discharge, adjusted by basic characteristics including male sex, age, and modified Charlson comorbidity index. Model 2: adjusted by variables of model 1 and resuscitation variables including arrest at home, witnessed, bystander cardiopulmonary resuscitation, shockable rhythm, cardiac cause, and arrest time. For a survival discharge, adjusted by variables of model 1 and resuscitation variables including arrest at home, witnessed, bystander cardiopulmonary resuscitation, shockable rhythm, cardiac cause, and arrest time. Model 3: adjusted by variables of model 2 and in-hospital variables including initial ECG, early CAG, PCI, echocardiography, SOFA score on day 1, and limitations of active treatment. For a survival discharge, adjusted by variables of model 2 and in-hospital variables including initial ECG, early CAG, PCI, echocardiography, ECMO, SOFA score, and limitations of active treatment.

**Table 3 jcm-12-05297-t003:** Univariate and multivariate logistic regression analyses for independent factors associated with a 6-month good neurological outcome (model 3).

	Univariate Analysis			Multivariate Analysis			Interaction Analysis		
	OR	95% CI	*p*	OR	95% CI	*p*	OR	95% CI	*p*
Baseline characteristics									
Age, per year	0.97	0.96–0.98	<0.001	0.96	0.95–0.98	<0.001	0.97	0.95–1.00	0.008
Men	1.66	1.26–2.17	<0.001	1.10	0.74–1.64	0.626	1.99	0.48–8.32	0.347
mCI									
mCI 0	Ref.								
mCI 1	0.54	0.40–0.73	<0.001	0.93	0.60–1.46	0.763	0.93	0.59–1.46	0.767
mCI 2	0.64	0.43–0.95	0.028	1.09	0.62–1.92	0.775	1.09	0.62–1.92	0.775
mCI 3	0.38	0.24–0.61	<0.001	1.25	0.64–2.42	0.516	1.23	0.64–2.39	0.534
Resuscitation variables									
Arrest at home	0.60	0.47–0.76	<0.001	0.87	0.62–1.23	0.440	0.88	0.62–1.24	0.456
Witnessed	3.09	2.30–4.16	<0.001	1.34	0.88–2.06	0.176	1.34	0.88–2.09	0.176
Bystander CPR	1.51	1.18–1.93	0.001	0.96	0.67–1.38	0.821	0.96	0.67–1.37	0.807
Shockable rhythm	12.86	9.75–16.95	<0.001	6.00	4.02–8.96	<0.001	6.01	4.02–8.97	<0.001
Cardiac etiology	7.33	5.23–10.14	<0.001	3.09	1.88–5.07	<0.001	3.09	1.88–5.07	<0.001
Downtime, min *	0.95	0.94–0.96	<0.001	0.94	0.93–0.95	<0.001	0.94	0.93–0.95	<0.001
SOFA score on day 1									
4–7	Ref								
8–12	0.30	0.21–0.42	<0.001	0.30	0.19–0.48	<0.001	0.30	0.19–0.48	<0.001
≥12	0.10	0.07–0.15	<0.001	0.18	0.11–0.30	<0.001	0.18	0.11–0.29	<0.001
STEMI or LBBB	1.87	1.38–2.55	<0.001	0.77	0.47–1.27	0.303	0.77	0.47–1.26	0.295
In-hospital process									
Early CAG	3.05	2.37–3.92	<0.001	1.38	0.87–2.20	0.166	1.38	0.87–2.18	0.174
Early PCI	2.92	2.11–4.05	<0.001	1.37	0.96–2.49	0.298	1.38	0.76–2.51	0.291
Early Echocardiography	1.44	1.12–1.85	0.004	1.03	0.71–1.49	0.870	1.03	0.71–1.49	0.878
ECMO	1.24	0.73–2.12	0.427						
Limiting active treatments	0.006	0.00–0.05	<0.001	0.01	0.00–0.08	<0.001	0.01	0.00–0.08	<0.001
Age × sex							0.99	0.96–1.01	0.401

* Arrest-to-ROSC interval. Abbreviations: mCI, modified Charlson comorbidity index; CPR, cardiopulmonary resuscitation; TTM, targeted temperature management; SOFA, Sequential Organ Failure Assessment; CAG, coronary angiography; STEMI, ST elevation myocardial infarction; LBBB, left bundle branch block; PCI, percutaneous coronary intervention; ECMO, extracorporeal membrane oxygenation.

**Table 4 jcm-12-05297-t004:** A good neurological outcome after targeted temperature management, stratified by sex and age.

	Age < 50 Years (Men = 260, Women = 131)	*p*	Age ≥ 50 Years (Men = 692, Women = 256)	*p*
Good neurological outcome, *n* (%)				
Women	41 (31.3)	0.009	50 (19.5)	0.002
Men	117 (45.0)		204 (29.5)	
for good neurologic outcome				
Crude OR (95% CI)	1.80 (1.15–2.80)	0.010	1.72 (1.21–2.44)	0.002
Adjusted OR * (95% CI)	1.49 (0.72–3.06)	0.279	1.05 (0.62–1.76)	0.866

* ORs are adjusted for modified Charlson comorbidity index, arrest at home, witnessed, bystander CPR, shockable rhythm, cardiac cause, arrest time, initial ECG, early CAG, PCI, echocardiography, ECMO, SOFA score, and limitations of active treatment. OR, odds ratio; CI, confidence interval.

**Table 5 jcm-12-05297-t005:** Associating factors with decision regarding limiting active treatment in patients who underwent multimodal prognostic tests according to resuscitation guidelines (*n* = 672).

	Univariate Analysis			Multivariate Analysis		
	OR	95% CI	*p*	OR	95% CI	*p*
Model including each single predictor						
Age, per year	1.02	1.00–1.03	0.013	1.02	1.00–1.03	0.029
Men	1.15	0.75–1.78	0.523	1.20	0.77–1.88	0.419
mCI						
mCI 0	Ref					
mCI 1	1.38	0.84–2.27	0.204	1.17	0.69–1.98	0.560
mCI 2	1.50	0.80–2.82	0.207	1.32	0.67–2.59	0.418
mCI 3	1.76	0.95–3.23	0.071	1.58	0.83–3.01	0.168
Absence of brainstem reflexes	2.17	1.44–3.27	<0.001	2.03	1.33–3.11	0.001
Injury on Brain CT or MRI	1.21	0.81–1.81	0.358			
Highly malignant EEG	1.04	0.66–1.63	0.880			
High NSE levels	1.25	0.84–1.87	0.274			
Absent SEP	2.10	1.30–3.38	0.002	1.95	1.19–3.21	0.009
Model based on international guideline						
Age, per year	1.02	1.00–1.03	0.013	1.02	1.00–1.03	0.033
Men	1.15	0.75–1.78	0.523	1.17	0.76–1.82	0.478
mCI						
mCI 0	Ref					
mCI 1	1.38	0.84–2.27	0.204	1.27	0.76–2.13	0.370
mCI 2	1.50	0.80–2.82	0.207	1.30	0.67–2.51	0.443
mCI 3	1.76	0.95–3.23	0.071	1.61	0.85–3.05	0.141
≥two of poor predictors	1.81	1.17–2.82	0.008	2.00	1.28–3.13	0.002

## Data Availability

The dataset analyzed in the current study is not publicly available because of contracts with the hospitals providing data to the database.

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
