# Peer review of "Sex Difference on Neurological Outcomes and Post-Cardiac Arrest Care in Out-of-Hospital Cardiac Arrest Patients Treated with Targeted Temperature Management: Post-Hoc Study of a Prospective, Multicenter, Observational Cohort Study"

_jcm, 2023, doi:10.3390/jcm12165297_

Round 1

Reviewer 1 Report

Thankyou for the opportunity to review you rmanuscript on sex diffferences for OHCA with TTM. 

Overall, this study is unique in its population, but less so in its clinical question. 

Introduction. 
Concise

Materials/methods.
Concise, thorough

Results. 
Comprehensive. Table 5 is very difficult to read and could be reformatted or added as a supplementary. 

Discussion. 
Well argued, good flow. Limitations covered off important points. 

Conclusion
Concise - although i would lead with your findings (ie no differences after adjusting for confounders). 

reasonable, minor grammar only

Author Response

We would like to thank you for this insightful comment. We did our best to revise the content as recommended. We strongly believe that these recommendations have enhanced the quality of the manuscript.

Table 5 is very difficult to read and could be reformatted or added as a supplementary. 

We motified Table 5 to Table S1 as your recommand. 

Reviewer 2 Report

I read with interest the manuscript on sex difference on neurological outcome in out-of-hospital cardiac arrest treated with TTM.  

It is well written, clear and interesting for readers. Methodology is correct, tables and figures are clear.

I have no major concerns.

I have only one minor suggestion:

-            At the end of introduction, please, specify the first appearance of TTM  

-            Please, add the recent reference of Ines Lakbar et al. published on Ann Intensive Care 2022 “Sex and out-of-hospital cardiac arrest survival: a systematic review”.

Author Response

We would like to thank you for this insightful comment. We did our best to revise the content as recommended. We strongly believe that these recommendations have enhanced the quality of the manuscript.

1. At the end of introduction, please, specify the first appearance of TTM  

We have edited it to "targeted temperature management (TTM)".

2. Please, add the recent reference of Ines Lakbar et al. published on Ann Intensive Care 2022 “Sex and out-of-hospital cardiac arrest survival: a systematic review”.

We have added the article as the 15th reference.